# U-Net Convolutional Neural Network for Real-Time Prediction of the Number of Cultured Corneal Endothelial Cells for Cellular Therapy

**DOI:** 10.3390/bioengineering11010071

**Published:** 2024-01-11

**Authors:** Naoki Okumura, Takeru Nishikawa, Chiaki Imafuku, Yuki Matsuoka, Yuna Miyawaki, Shinichi Kadowaki, Makiko Nakahara, Yasushi Matsuoka, Noriko Koizumi

**Affiliations:** 1Department of Biomedical Engineering, Faculty of Life and Medical Sciences, Doshisha University, 1-3 Miyakodani, Tatara, Kyotanabe-City 610-0394, Kyoto, Japan; takeru_nishikawa@icloud.com (T.N.); yuki.m.99@icloud.com (Y.M.); cof.green617@gmail.com (Y.M.); jt-jjc38@mail.doshisha.ac.jp (S.K.); nkoizumi@mail.doshisha.ac.jp (N.K.); 2ActualEyes Inc., D-egg, 1 Jizodani, Koudo, Kyotanabe-City 610-0332, Kyoto, Japan; makiko.nakahara@actualeyes.co.jp (M.N.); yasushi.matsuoka@actualeyes.co.jp (Y.M.)

**Keywords:** corneal endothelial cell, tissue engineering, cellular therapy, artificial intelligence, deep learning, U-Net

## Abstract

Corneal endothelial decompensation is treated by the corneal transplantation of donor corneas, but donor shortages and other problems associated with corneal transplantation have prompted investigations into tissue engineering therapies. For clinical use, cells used in tissue engineering must undergo strict quality control to ensure their safety and efficacy. In addition, efficient cell manufacturing processes are needed to make cell therapy a sustainable standard procedure with an acceptable economic burden. In this study, we obtained 3098 phase contrast images of cultured human corneal endothelial cells (HCECs). We labeled the images using semi-supervised learning and then trained a model that predicted the cell centers with a precision of 95.1%, a recall of 92.3%, and an F-value of 93.4%. The cell density calculated by the model showed a very strong correlation with the ground truth (Pearson’s correlation coefficient = 0.97, *p* value = 8.10 × 10^−52^). The total cell numbers calculated by our model based on phase contrast images were close to the numbers calculated using a hemocytometer through passages 1 to 4. Our findings confirm the feasibility of using artificial intelligence-assisted quality control assessments in the field of regenerative medicine.

## 1. Introduction

Corneal endothelial decompensation is commonly addressed through corneal transplantation using donor corneas. In recent decades, lamellar posterior procedures, such as endothelial transplants, including Descemet’s stripping automated endothelial keratoplasty (DSAEK) and Descemet’s membrane endothelial keratoplasty (DMEK), have gained increasing popularity worldwide [1,2,3,4]. These procedures have experienced rapid growth in the U.S. and Europe and now constitute approximately 60% and 50%, respectively, of all corneal grafting surgeries in these regions, although adoption in the rest of the world has been slower. These procedures, on the other hand, are technically challenging and may present complications, such as the detachment of the graft or haze between layers, and they may be significantly more difficult to perform in complex cases (e.g., aphakic eyes or eyes with a past history of filtration surgery or posterior vitrectomy) [5,6,7,8]. A further significant issue is the shortage of donor corneas—at present, worldwide, only one in seventy patients requiring corneal transplantation are able to receive a donor cornea [9]. Consequently, many researchers have looked to regenerative medicine as a means of providing more efficient treatments and overcoming the donor shortage [10,11,12,13,14,15].

In 2013, we initiated the first-in-human clinical research into cell therapy for corneal endothelial decompensation through a process that involves injecting cultured human corneal endothelial cells (HCECs), together with a ROCK inhibitor, into the anterior chamber of the eye [16]. All of the first 11 cases associated with that study recovered corneal transparency with the restoration of an in vivo-like sheet structure of the corneal endothelium at the posterior side of the cornea. The five-year results showed that 10 of the 11 patients maintained their corneal transparency with no adverse side effects, such as rejection, irreversible glaucoma, or tumor formation [17]. However, challenges in the efficient manufacturing of HCECs for clinical use still limit the efficacy of tissue engineering protocols.

Cell culturing for general clinical use typically involves passaging for in vitro expansion at designated intervals, and the total number of cultured cells is then estimated by manual counting with a cell counter or hemocytometer when the cells are passaged, meaning that cell growth is evaluated only at passaging. We were motivated to develop a real-time monitoring system for HCEC growth to facilitate more efficient cell manufacturing. We hypothesized that utilizing an artificial intelligence (AI) model for the image segmentation of cultured HCECs would enable the real-time and nondestructive prediction of the total cell numbers throughout the manufacturing process.

## 2. Materials and Methods

### 2.1. Ethics Statement

The human tissue used in this study was handled following guidelines based on the ethical principles of the Declaration of Helsinki. Normal human donor corneas were obtained from Eversight (Ann Arbor, MI, USA). No tissues were procured from prisoners. HCECs were cultured according to a protocol approved by the ethical review committee of ActualEyes Inc. (Approval No. 22-01).

### 2.2. Cell Cultures

Donor corneas were stored at 4 °C in a storage medium (Optisol-GS; Chiron Vision, Irvine, CA, USA) for fewer than 20 days before use in corneal endothelial cell cultures. Cultured HCECs from 54 paired corneas from 27 donors (age range from 5 to 60 years old) were used for the generation of AI models. An additional 5 paired corneas from 5 donors (ages 30, 33, 34, 35, and 37 years old) were utilized to culture HCECs, and the cell density and total numbers of HCECs were estimated by the generated AI models to evaluate the feasibility of the models.

Descemet’s membrane, including the corneal endothelium, was stripped from the donor corneas using an IOL SINSKEY curved hook (Inami & Co., Ltd., Tokyo, Japan). The HCECs were cultured according to a previously reported protocol [18]. Briefly, Descemet’s membranes from paired corneas (2 corneas) were incubated in OptiMEM-I (Thermo Fisher Scientific, Waltham, MA, USA) supplemented with 1 mg/mL collagenase A (Roche Applied Science, Penzberg, Germany) and digested at 37 °C for 16 h. The released HCECs were washed with OptiMEM-I and suspended in a culture medium composed of OptiMEM-I, 8% fetal bovine serum, 5 ng/mL epidermal growth factor (EGF; Thermo Fisher Scientific), 10 μM of SB203580 (Cayman Chemical, Ann Arbor, MI, USA), 20 μg/mL ascorbic acid (Sigma-Aldrich, St. Louis, MO, USA), 200 mg/L calcium chloride, 0.08% chondroitin sulfate (Sigma-Aldrich), and 50 μg/mL gentamicin (Thermo Fisher Scientific). The HCECs were then seeded in one well of a 6-well culture plate and cultured in a humidified atmosphere at 37 °C in 5% CO_2_. The culture medium was replaced with fresh medium 3 times a week. The HCECs were passaged at a 1:3 ratio every 10 days until passage 4 by trypsinization with 0.05% Trypsin-EDTA (Thermo Fisher Scientific). Phase contrast images (1392 × 1040 pixels) of the cultured HCECs were obtained using DM14000B (Leica Microsystems, Inc., Wetzlar, Germany) at the time of every passage (after 10 days of primary culture and at each passage). The total cell numbers were calculated using a hemocytometer at every passage.

### 2.3. A Flow Chart for Generating the Models for Predicting Cell Centers

A flow chart for generating the “initial model” and the “cell count model” is shown in Figure 1. From a total of 3098 phase contrast images, 606 images were used as the dataset for creating the “initial model” as ground truth. The center of each cell was annotated manually and confirmed by three researchers. A discussion was conducted to reach a consensus for cells that were difficult to annotate. The 606 images were divided into three categories (training data 1: 402 images; validation data 1: 121 images; and test data: 83 images). Preprocessing was performed using Python3 (Python Software Foundation, https://www.python.org/ (accessed on 29 December 2023), Beaverton, OR, USA) and the Python image processing library OpenCV. The images were then resized by a factor of 1/4 (348 × 360 pixels) to prevent memory overflow on the Graphics Processing Unit (GPU) used for training. Gamma correction, contrast adjustment, and highlighting of the cell boundaries were also conducted. Histogram flattening was then used to flatten the pixel values and remove fine noise. Training dataset 1 was augmented by inverting the images.

### 2.4. Generation of the “Initial Model”

The “initial model” used to predict the center of the HCECs was generated by U-Net. The program was created using Python 3 and Chainer (Preferred Networks, Inc., Tokyo, Japan). The ReLU function [19] was used as the activation function for deep learning, and Adam [20] was used as the optimization method. The optimal number of training cycles was determined using the value of the cross-entropy error. The network was generated by learning 402 images (training data 1) for up to 30 epochs, and then, the network was validated using 121 images (validation data 1).

### 2.5. Generation of the “Cell Count Model” Using Semi-Supervised Learning

The “initial model” was used to annotate the 2492 remaining unlabeled images. Plot omissions and over-detections in the predicted images were then manually revised using the iPad drawing tool Procreate (version 5.0.2 for iPadOS; Savage Interactive Pty Ltd., Vienna, Austria) to create ground truth. The 3098 ground truths were divided into training data 2 (n = 2341) and validation data 2 (n = 757). Data augmentation of training data 2 was performed on the training data to create 9364 images.

The “cell count model” was generated by U-Net by applying the same method used to generate the “initial model.” The network was generated by learning 2341 images (training data 2) for up to 30 epochs, and then, the network was validated using 757 images (validation data 2). Finally, the cell centers of manually annotated test data (n = 83) were predicted by the “initial model” and the “cell count model”, and the performance was evaluated using the parameters of precision, recall, and F-measure.

### 2.6. Feasibility of Using the “Cell Count Model” to Predict Total Cell Numbers in Cultures

Five paired corneas were cultured and passaged 4 times. The total cell numbers in culture dishes or flasks were determined using a hemacytometer in every passage. The cell centers of phase contrast images obtained in every passage were predicted by the “initial model” and the “cell count model”, and the cell densities in each image were calculated. The total cell numbers in the culture dishes or flasks were then calculated based on the cell densities predicted by both models.

## 3. Results

The “initial model” was generated by the supervised learning of 402 manually annotated images of training dataset 1 utilizing U-Net. To reduce the time required to prepare larger numbers of ground truth images, we generated the “initial model” as a first step to annotate the remaining 2492 unlabeled images. The validation loss showed a low value through 10–14 epochs and gradually increased after 15 epochs, while the training loss continuously decreased throughout 30 epochs, indicating overfitting after 15 epochs. We then selected the model generated by 14 epochs as the “initial model” (Figure 2A). Data annotated using the “initial model” were then manually corrected by the researchers as a second step to generate ground truth. The time required for manual annotation to prepare ground truth was 1216.0 ± 81.0 s/image, while the time needed for annotation by the “initial model,” followed by manual correction, was 40.7 ± 20.0 s/image.

We then generated the “cell count model” using a total of 3098 annotated images based on the method of semi-supervised learning. The validation loss decreased and became almost stable after 6 epochs, although the training loss continuously decreased throughout the 30 epochs. To avoid overfitting, the model generated by six epochs was selected as the “cell count model” (Figure 2B). Representative phase contrast images, with predicted cell centers shown in pink, indicated that both the “initial model” and the “cell count model” succeeded in predicting the cell centers of the HCECs (Figure 3). The cell density was occasionally low in some cultures; therefore, we confirmed whether our models still worked well in cultures with various cell densities. Representative predicted images of low-, middle-, and high-cell-density cultures showed that the cell centers were well predicted by both models in all cultures.

The average prediction time for the “cell count model” was 5.8 ± 0.0 s/page. The performance of the models for predicting cell centers was evaluated by calculating precision, recall, and F-value. The precision, recall, and F-value were 95.4%, 89.3%, and 91.9%, respectively, for the “initial model” and 95.1%, 92.3%, and 93.4%, respectively, for the “cell count model” (Table 1). The precision of the “cell count model” was similar to that of the “initial model”, but the recall and F-value were higher for the “cell count model” than for the “initial model”, indicating that the performance of the “cell count model” had been improved by learning larger numbers of data.

We used the predicted cell centers in the phase contrast images of cultured HCECs to calculate the cell density. When calculated using the “initial model”, the cell density showed a very strong correlation with the ground truth (Pearson’s correlation coefficient = 0.97, *p* value = 7.07 × 10^−50^). Similarly, the cell density calculated using the “cell count model” also showed a very strong correlation with the ground truth (Pearson’s correlation coefficient = 0.97, *p* value = 8.10 × 10^−52^). The Pearson’s correlation coefficients of the associations between the “initial model” and the ground truth and between the “cell count model” and the ground truth were at the same level (Figure 4).

We also cultured five lots of HCECs derived from five donors to evaluate the feasibility of using the “cell count model” clinically to predict the total cell numbers (Figure 5). The blue color in the graphs in Figure 5 indicates the total cell numbers calculated manually using a hemocytometer. The total cell numbers were also determined by the cell density calculated based on the cell centers predicted by our models. Both the “initial model” and the “cell count model” successfully predicted the total cell numbers from passage 1 to 4 in all five lots. All phase contrast images of all five lots in all passages were successfully predicted by both models. The “cell count model”, rather than the “initial model”, tended to show a closer prediction to the manual hemacytometer count, especially in passage 4 (Figure 5).

## 4. Discussion 

In this study, we trained U-Net to develop an AI model to predict cell density based on phase contrast images. This model will eventually enable the real-time and nondestructive calculation of total cell numbers throughout the cell manufacturing process. Here, we obtained 3098 phase contrast images of cultured HCECs during primary culture (passage 0) and up to passage 4. Large numbers of data usually allow the generation of high-performance AI; however, manual data annotation is particularly labor-intensive. In the current study, the manual annotation of the cell centers in the phase contrast images by the researchers took 1216.0 ± 81.0 s/image, implying that annotating all 3098 images would take approximately 1050 h. Therefore, we utilized semi-supervised learning to reduce the labor-dependent labeling process.

We first generated the “initial model” from 606 labeled data, which were manually annotated by researchers. This “initial model” was then used to annotate the remaining 2492 unlabeled data. Our subsequent training of the final model (the “cell count model”) using semi-supervised learning further increased the performance above that of the “initial model”. Our feasibility study using five independent HCEC lots demonstrated that the generated model succeeded in predicting total cell numbers from phase contrast images.

Semi-supervised learning is a machine learning paradigm that uses both labeled and unlabeled data for training [21]. The primary motivation for choosing semi-supervised learning is that labeling the data is time-consuming and/or expensive, whereas obtaining unlabeled data requires little time or expense. Semi-supervised learning is widely used in multiple research fields, including the medical field, where it is applied, for example, in oncology diagnostics and care [22,23]. The classification of histopathology and radiotherapy images is essential for various kinds of cancers, such as breast, lung, gastric, liver, colorectal, kidney, pancreatic, and uterine cancers [24,25,26,27,28,29], but adequate labeling by experts is often time-consuming and thus cost-ineffective. Semi-supervised learning is a powerful tool for dealing with the dilemma of big data, especially in the context of images that are usually acquired in abundance during routine clinical flows. Therefore, we were motivated to utilize semi-supervised learning in the current study because we routinely obtained phase contrast images during the cell manufacturing process for our records, but labeling is not performed routinely because it is so time-consuming. We expected to generate more accurate models by training using semi-supervised learning and a combination of both labeled and unlabeled data than by training using only labeled data [21]. Indeed, in this study, the F-value of the “cell count model” was increased from 91.9% to 93.4% by the “initial model”. Our data suggest that semi-supervised learning has applications in the tissue engineering field for evaluating the status of in vitro cells because phase contrast images are routinely acquired in abundance during cell manufacturing processes.

In the present study, we trained our models using U-Net, which is a convolutional neural network (CNN) developed by Ronneberger and colleagues for biomedical image segmentation [30]. U-Net consists of a contracting path, which is similar to a typical CNN and an expanding path and includes an upsampling of the feature map followed by a 2 × 2 convolution. The strong use of data augmentation to utilize the available annotated data enables efficiency and effectiveness in various image segmentation processes with a small amount of training data. The first analysis of the corneal endothelium, conducted by Fabijańska, determined that a U-Net-based CNN can perform the segmentation of HCECs based on the specular microscopy images and reported an Area Under the Receiver Operating Characteristic Curve (AUROC) level of 0.92 [31]. Subsequent research also showed that U-Net can be applied to segment HCECs for the evaluation of healthy or diseased corneal endothelia [32,33,34] and post-endothelial corneal keratoplasty based on specular microscopy images [35]. The limitation of our study is that we only utilized U-Net, which is still considered a foundational model, especially in medical imaging, although the field of AI for image segmentation is rapidly evolving [36,37,38]. Future continuous efforts using newer models are anticipated to offer improved performance, adaptability, and efficiency.

We previously achieved the segmentation of CECs with the presence of guttae in a Fuchs endothelial corneal dystrophy model mouse [39]. Here, we expanded the use of U-Net from the analysis of in vivo cells to the analysis of in vitro cells for the purpose of quality control in the context of regenerative medicine. Our model predicted the total cell numbers; therefore, it can provide real-time data for the numbers of manufactured cells generated throughout the culture period. Manufacturers may utilize our proposed model to optimize the timing of cell passages, seeding cell density, and the supplementation of the components of the culture medium [40,41]. Although those parameters are usually fixed depending on the cell type, future optimization of these parameters may maximize the efficiency of cell manufacture. A potential limitation in the context of using this model for manufacturing clinical use cells is that deciding those parameters based on the AI model might require challenging communication with regulatory authorities. Collecting data showing the equivalence of manually counted cell numbers (i.e., the conventional method) and AI-predicted cell numbers is necessary before a company utilizes our model in its cell manufacturing process.

Our research group is developing a cryopreserved corneal endothelial cellular product and is currently in consultation with the Japanese regulatory authority to start clinical trials. For clinical use, strict quality control assessments are necessary during the manufacture of these cells to ensure their safety and efficacy [42,43]. In addition, the efficiency of cell manufacturing should be optimized to enable cell therapy to be a sustainable standard procedure with an acceptable economic burden. Here, we have shown the feasibility of using AI-assisted quality control assessments for corneal endothelial cell therapy; however, this method can theoretically be utilized in various cell types for other fields of regenerative medicine.

## Figures and Tables

**Figure 1 bioengineering-11-00071-f001:**
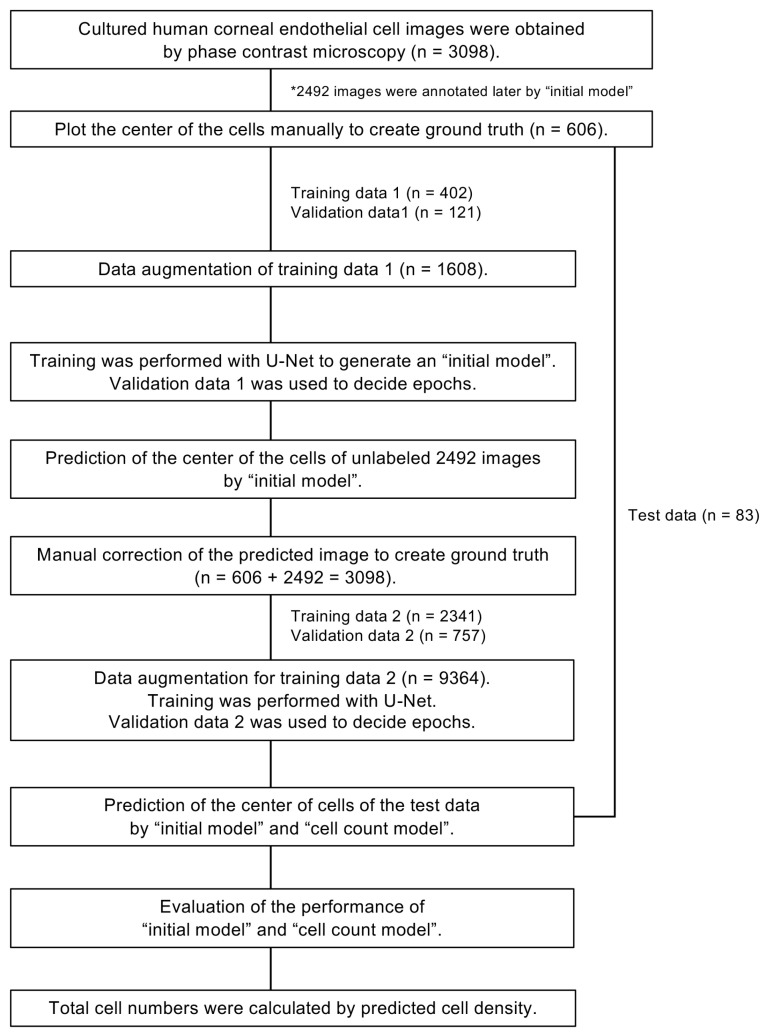
Flowchart showing the generation of AI for predicting the centers of cultured human corneal endothelial cells based on their phase contrast images. In total, 3098 phase contrast images of cultured human corneal endothelial cells (HCECs) were obtained. Of those, 606 images were manually annotated and used as the dataset for creating the “initial model” as ground truth. The 606 images were divided into three categories (training data 1: 402 images; validation data 1: 121 images; and test data: 83 images). Training data 1 was used to generate the “initial model” using U-Net. The remaining unannotated 2492 images were then annotated using the “initial model”, followed by manual correction. A total of 3098 ground truth images were divided into training data 2 (n = 2341) and validation data 2 (n = 757). The “cell count model” was generated using U-Net. Finally, the performance was evaluated using test data.

**Figure 2 bioengineering-11-00071-f002:**
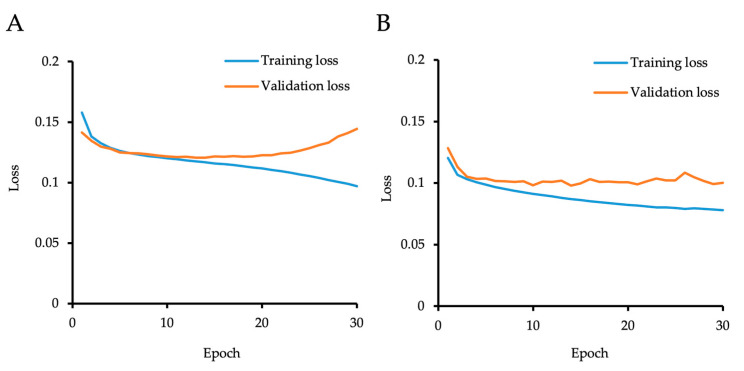
The training and validation loss curves of the networks. (**A**) The network was generated by learning 402 images (training data 1) for up to 30 epochs, and the network was validated using 121 images (validation data 1). The validation loss showed a low value through 10–14 epochs and gradually increased after 15 epochs, while the training loss continuously decreased throughout 30 epochs. The model generated by 14 epochs was selected as the “initial model”. (**B**) The network was generated by learning 2341 images (training data 2) for up to 30 epochs, and then, the network was validated using 757 images (validation data 2). The validation loss decreased and became almost stable after 6 epochs, though training loss continuously decreased throughout 30 epochs. The model generated by 6 epochs was selected as the “cell count model”.

**Figure 3 bioengineering-11-00071-f003:**
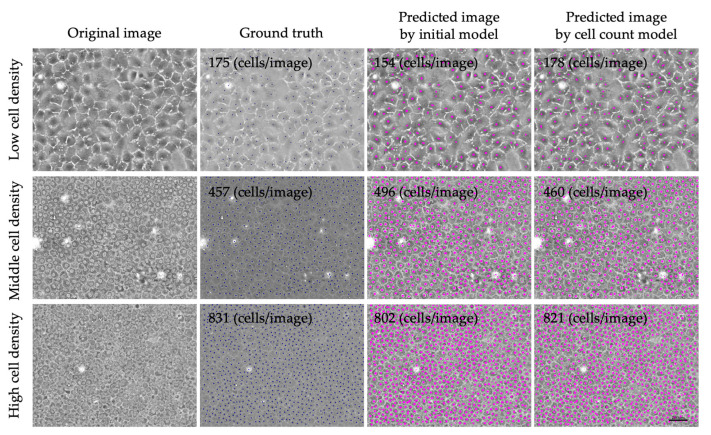
Representative phase contrast images with prediction of the cell centers using the “initial model” and the “cell count model”. Cell centers of cultured human corneal endothelial cells were well predicted by both AIs at various cell densities. Blue dots in the ground truth images indicate the manually annotated cell center. Pink dots indicate the cell centers predicted by the AIs. Scale bar: 200 μm.

**Figure 4 bioengineering-11-00071-f004:**
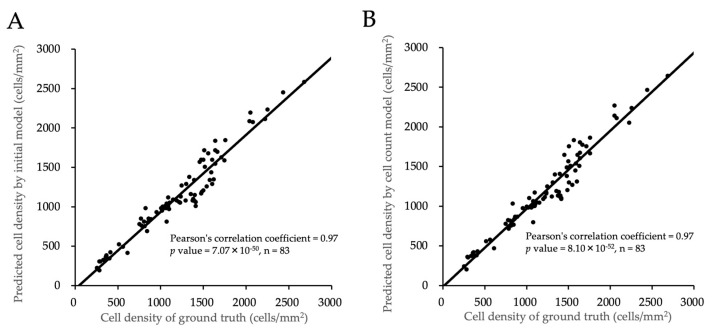
Correlation between the cell density predicted by generated AIs and the ground truth. (**A**) The cell density calculated by “initial model” shows very strong correlation with the ground truth (Pearson’s correlation coefficient = 0.97, *p* value = 7.07 × 10^−50^). In total, 83 test data were evaluated using the “initial model”. (**B**) The cell density calculated by the “cell count model” also shows very strong correlation with the ground truth (Pearson’s correlation coefficient = 0.97, *p* value = 8.10 × 10^−52^). In total, 83 test data were evaluated using the “cell count model”.

**Figure 5 bioengineering-11-00071-f005:**
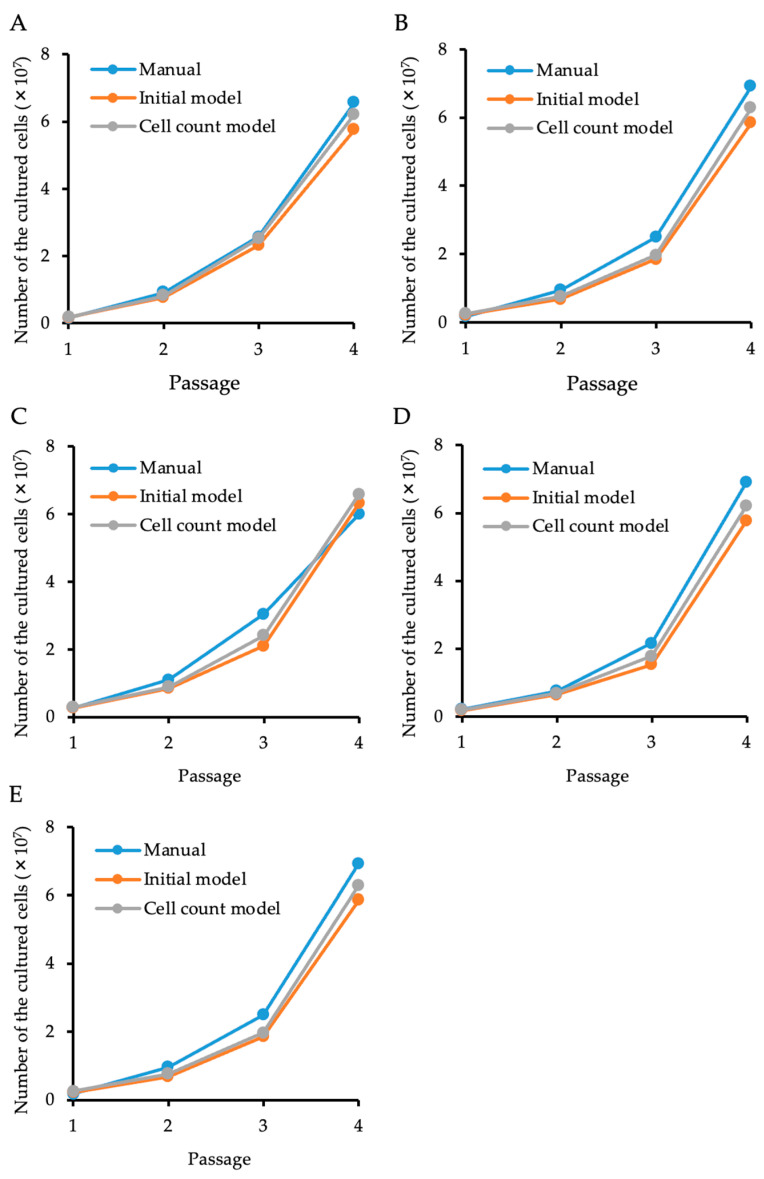
Total cell numbers at the time of every culture passage, as predicted by generated AIs. Five paired corneas were cultured and passaged 4 times. Total cell numbers were determined using a hemacytometer at the time of every culture passage. Cell centers in phase contrast images were predicted by the “initial model” and the “cell count model”, and the total cell numbers in the culture dish or flask were calculated. Both AIs successfully predicted the total cell numbers from passages 1 to 4 in all 5 lots, but a closer prediction to a manual count tended to be observed with the “cell count model” rather than with the “initial model”. Figure (**A**–**E**) show the results of 5 independent cell lots.

**Table 1 bioengineering-11-00071-t001:** Precision, recall, and F-value of the “initial model” and “cell count model”.

	Precision (%) (95% CI)	Recall (%) (95% CI)	F-Value (%) (95% CI)
Initial model	95.4 (94.4–96.3)	89.3 (87.5–91.0)	91.9 (90.9–92.9)
Cell count model	95.1 (94.1–96.1)	92.3 (90.8–93.8)	93.4 (92.6–94.2)

## Data Availability

Data are contained within the article.

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
