# Peer review of "U-Net Convolutional Neural Network for Real-Time Prediction of the Number of Cultured Corneal Endothelial Cells for Cellular Therapy"

_bioengineering, 2024, doi:10.3390/bioengineering11010071_

Round 1
Reviewer 1 Report (Previous Reviewer 1)
Comments and Suggestions for Authors
The authors addressed all suggested points.
Comments on the Quality of English LanguageThe English was improved.
Reviewer 2 Report (Previous Reviewer 2)
Comments and Suggestions for Authors
Interesting study. All previously highlighted issues have been appropriately addressed.
This manuscript is a resubmission of an earlier submission. The following is a list of the peer review reports and author responses from that submission.
Round 1
Reviewer 1 Report
Comments and Suggestions for Authors
After reviewing the manuscript titled "U-Net Convolutional Neural Network for Real-time Prediction of the Number of Cultured Corneal Endothelial Cells for Cellular Therapy," I have compiled a list of detailed suggestions for revision:
- Clarity and Structure:
o The manuscript is well-structured and presents a clear narrative. However, some sections, particularly the Materials and Methods, could benefit from additional subheadings for better readability and organization.
o The introduction provides a good background but could be enhanced by more explicitly stating the research question or hypothesis.
- Technical Content and Depth:
o The technical description of the U-Net Convolutional Neural Network is comprehensive but could be made more accessible to readers not familiar with deep learning. A brief introductory explanation of key concepts would be beneficial.
o More detailed discussion on the choice of parameters and the configuration of the neural network would enhance the reader's understanding of the model's design.
- Methodology:
o The Materials and Methods section is thorough, but the study would benefit from a more detailed description of the validation process for the AI models.
o A comparative analysis with other existing methods or models, if available, would strengthen the manuscript by highlighting the novel contributions of this study.
- Results and Data Interpretation:
o The Results section is well-detailed, but the manuscript could benefit from a more in-depth discussion of any anomalies or unexpected findings in the data.
o Additional visualizations, such as graphs or more detailed tables, could help in better illustrating the findings, particularly the performance metrics of the AI models.
- Discussion:
o The Discussion section effectively ties the results back to the broader context, but it could delve deeper into the implications of the findings for the field of regenerative medicine.
o Discuss potential limitations of the study and how they might impact the interpretation of the results or the applicability of the AI models.
- References and Citations:
o Ensure that all references are current and relevant. Adding more recent studies, particularly from 2023 and 2024, would strengthen the manuscript's position within current research trends.
o Consider adding more recent references to support statements, especially in rapidly evolving areas of the field. Where possible, include recent studies to demonstrate the manuscript's alignment with current research trends. In particular, consider including additional references to support the discussion and to provide context to the study’s findings. I suggest adding data related to recent bulk transcriptomics studies which could represent a strong substrate to enforce the role of described molecular mechanisms, such as the recent PMID: 26115622 and PMID: 32184807.
o Check for consistency in citation format and correct any discrepancies.
- Figures and Tables:
o Ensure that all figures and tables are clearly labeled and have descriptive legends. Consider adding more figures or charts to visually represent key findings.
o Verify the quality and resolution of all images to ensure they are clear and easily interpretable.
- Ethical Considerations and Conflicts of Interest:
o The manuscript addresses ethical considerations well. However, ensure that any potential conflicts of interest, especially with the involvement of ActualEyes Inc., are transparently disclosed.
- Conclusion and Future Work:
o The conclusion effectively summarizes the findings, but it could also suggest potential areas for future research or applications of the AI models in other contexts.
o Consider discussing how this research could pave the way for further innovations in cellular therapy and regenerative medicine.
Overall, the manuscript presents valuable research with clear implications for the field of regenerative medicine. These revisions should help to enhance its clarity, depth, and scientific rigor.
Comments on the Quality of English LanguageThe English should be improved, especially in technical paragraphs.
Reviewer 2 Report
Comments and Suggestions for Authors
Line 28. It reads: “Corneal endothelial decompensation is typically treated by corneal transplantation of donor corneas, and endothelial transplantation accounts for approximately 40% of all corneal transplants [1]. Modern endothelial keratoplasties, such as Descemet's stripping automated endothelial keratoplasty (DSAEK) and Descemet's membrane endothelial keratoplasty (DMEK), have become increasingly popular treatments for corneal endothelial decompensation [2,3].”
Comment: The study by Gain et al analyzed data gathered between 2012 and 2013, and reported that around 30% were lamellar (including both anterior and posterior). The second cited article by Tan et al. was published in 2012. Therefore, they are rather outdated.
Therefore, consider modifying to: “Corneal endothelial decompensation is commonly addressed through corneal transplantation using donor corneas. In recent decades, lamellar posterior procedures, such as endothelial transplants, including Descemet's stripping automated endothelial keratoplasty (DSAEK) and Descemet's membrane endothelial keratoplasty (DMEK)), have gained increasing popularity worldwide. These procedures have experienced rapid growth in the USA and Europe, constituting approximately 60% and 50% of all corneal grafting surgeries in these regions, respectively. Though, adoption in the rest of the world has been slower.”
References to be cited:
Chaurasia S, Mohamed A, Garg P, et al. Thirty years of eye bank experience at a single centre in India. Int Ophthalmol 2020;40:81–8.
Gao H, Huang T, Pan Z, et al. Survey report on keratoplasty in China: a 5- year review from 2014 to 2018. Plos one 2020;15:e0239939
Flockerzi E, Turner C, Seitz B, Collaborators GSG; GeKeR Study Group. Descemet's membrane endothelial keratoplasty is the predominant keratoplasty procedure in Germany since 2016: a report of the DOG-section cornea and its keratoplasty registry [published online ahead of print, 2023 Aug 16]. Br J Ophthalmol. 2023;bjo-2022-323162. doi:10.1136/bjo-2022-323162
Eye Bank Association of America. Statistical Report. https://restoresight.org/members/publications/statistical-report/
Line 33. It reads: “However, these procedures have several problems, such as detachment of the graft from the recipient's cornea, haze between layers, and difficulties in surgery for complicated cases (e.g., eyes with filtration surgery, vitrectomy, or iris defect)”
Comment:
Consider modifying to: “These procedures are, on the other hand, technically challenging, and may present complications, such as detachment of the graft, haze between layers, and may be significantly more difficult in complex cases (e.g., aphakic eyes, or with past history of filtration surgery, or posterior vitrectomy)”
References to be cited:
Quilendrino R, Rodriguez-Calvo de Mora M, Baydoun L, et al. Prevention and Management of Descemet Membrane Endothelial Keratoplasty Complications. Cornea. 2017;36(9):1089-1095. doi:10.1097/ICO.0000000000001262
Berrospi RD, Galvis V, Bhogal M, Tello A. Double-Line Reflection Pattern as a Simple Method to Determine Graft Orientation of Descemet Membrane Endothelial Keratoplasty. Cornea. 2019;38(6):768-771. doi:10.1097/ICO.0000000000001889
Karadag R, Aykut V, Esen F, Oguz H, Demirok A. Descemet's membrane endothelial keratoplasty in aphakic and vitrectomized eye. GMS Ophthalmol Cases. 2020 Feb 14;10:Doc02. doi: 10.3205/oc000129. PMID: 32158637; PMCID: PMC7047886.
Aravena C, Yu F, Deng SX. Outcomes of Descemet Membrane Endothelial Keratoplasty in Patients With Previous Glaucoma Surgery. Cornea. 2017 Mar;36(3):284-289. doi: 10.1097/ICO.0000000000001095. PMID: 27893525; PMCID: PMC5290193.
Line 207. In the “Discussion” section, before the description of the number of phase contrast images obtained, I consider than a short introductory paragraph could be helpful for the reader. Something like: “This kind of artificial intelligence approach has been employed for the automatic segmentation of specular microscopy images of the corneal endothelium, experimentally and in humns, both in normal corneas and in cases of Fuchs' dystrophy. In this study, we applied the U-Net Convolutional Neural Network to cultured human corneal endothelial cells (HCECs) in order to determine cell density.”
Additional references:
Sierra J. S., Pineda J., Viteri E., Tello A., Millán M. S., Galvis V., Romero L. A., Marrugo A. G., “Generating density maps for convolutional neural network-based cell counting in specular microscopy images,” J. Phys.: Conf. Ser. 1547(1), 012019 (2020). 10.1088/1742-6596/1547/1/012019
Okumura, N.; Yamada, S.; Nishikawa, T.; Narimoto, K.; Okamura, K.; Izumi, A.; Hiwa, S.; Hiroyasu, T.; Koizumi, N. U-Net 356 Convolutional Neural Network for Segmenting the Corneal Endothelium in a Mouse Model of Fuchs Endothelial Corneal Dys-357 trophy. Cornea 2022, 41, 901-907, doi:10.1097/ico.0000000000002956.
Fabijańska A. Segmentation of corneal endothelium images using a U-Net-based convolutional neural network. Artif Intell Med. 2018 Jun;88:1-13. doi: 10.1016/j.artmed.2018.04.004. Epub 2018 Apr 19. PMID: 29680687.
Vigueras-Guillén JP, Sari B, Goes SF, Lemij HG, van Rooij J, Vermeer KA, van Vliet LJ. Fully convolutional architecture vs sliding-window CNN for corneal endothelium cell segmentation. BMC Biomed Eng. 2019 Jan 30;1:4. doi: 10.1186/s42490-019-0003-2. PMID: 32903308; PMCID: PMC7412678.
